# FALCON: FEEDBACK-DRIVEN ADAPTIVE LONG/SHORT-TERM MEMORY REINFORCED CODING OPTIMIZATION

## ABSTRACT

Recently, large language models (LLMs) have achieved significant progress in automated code generation. Despite their strong instruction-following capabilities, these models frequently struggled to align with user intent in the coding scenario. In particular, they were hampered by datasets that lacked diversity and failed to address specialized tasks or edge cases. Furthermore, challenges in supervised fine-tuning (SFT) and reinforcement learning from human feedback (RLHF) led to failures in generating precise, human-intent-aligned code. To tackle these challenges and improve the code generation performance for automated programming systems, we propose **F**eedback-driven **A**daptive **L**ong/short-term memory reinforced **C**oding **O**ptimizatio**N** (i.e., FALCON). FALCON is structured into two hierarchical levels, from the global level, long-term memory improves code quality by retaining and applying learned knowledge, while from the local level, short-term memory allows for the incorporation of immediate feedback from compilers and AI systems. Additionally, we introduce meta-reinforcement learning with feedback rewards to solve the global-local bi-level optimization problem and enhance the model's adaptability across diverse code generation tasks. Extensive experiments are conducted and it is found that our technique achieves state-of-the-art performance, leading other reinforcement learning methods by more than 4.5 percentage points on the MBPP benchmark and 6.1 percentage points on the Humaneval benchmark. The open-sourced code is publicly available at `https://anonymous.4open.science/r/FALCON-BFE0/README.md`.

## 1 INTRODUCTION

The development of Large Language Models (LLMs) has significantly advanced automated code generation (Zan et al., 2023). Models like CodeLLaMA (Roziere et al., 2023) and DeepSeek-Coder (Guo et al., 2024), tailored for code-centric tasks, have demonstrated outstanding performance across programming challenges. While LLMs excel in instruction-following through tuning (Jiang et al., 2024), they often misalign with user intent, making feedback-based adjustments critical. For example, InstructGPT (Ouyang et al., 2022) leverages reinforcement learning with human feedback (RLHF), and CodeRL (Le et al., 2022) uses compilation feedback to refine model performance. Similarly, CompCoder (Wang et al., 2022) enhances code compilability with compiler feedback, and RLTF (Liu et al., 2023a) offers fine-grained feedback on compiler errors. However, current RL frameworks generate compilation errors and overlook non-differentiable features (e.g. coding style) that affect the performance significantly (Jiang et al., 2024). To address these challenges, we propose a reinforcement learning system combining long-term and short-term memory feedback. From the global level, long-term memory tracks trends over time for higher-quality code retrieval, while from the local level, short-term memory captures recent errors and immediate feedback. The main contributions of this paper are as follows:

- **Short-Term and Long-Term Memory for Reinforcement Learning**: We propose a dual-memory approach for reinforcement learning in code generation, where short-term memory enables real-time corrections and long-term memory accumulates knowledge from past runs to improve code quality and reduce repetitive mistakes.

- **Non-Differentiable Code Features into Feedback Loops**: Our approach addresses the limitation of current RL frameworks by integrating non-differentiable code features like style, readability, and best practices into the feedback loop, ensuring the generated code is both functionally sound and aligned with real-world programming standards.

- **Meta-Reinforcement Learning for Generalization Across Tasks**: We enhance the model's versatility by incorporating meta-reinforcement learning, allowing it to efficiently generalize across diverse programming tasks, adapt quickly to new environments, and handle a wide range of coding challenges with fewer training iterations.

## 2 RELATED WORKS

### 2.1 PRE-TRAINED MODELS FOR CODE GENERATION

In recent years, pre-trained language models have made significant progress in the field of code generation. Trained on large-scale code corpora, these models have demonstrated powerful code generation capabilities. For example, CodeBERT Feng et al. (2020), a model based on an encoder-only architecture, has shown impressive results. With the advent of in-context learning methods, decoder-only Transformer models have become the dominant technology for language modeling Vaswani et al. (2017). Several models, such as CodeGPT Lu et al. (2021), CodeGeeX Zheng et al. (2024), and DeepSeek-Coder Guo et al. (2024), use Causal Language Modelling (CLM) pretraining, while others like CodeT5 Wang et al. (2021) and AlphaCode Li et al. (2022) utilize an encoder-decoder architecture. Additionally, models like CodeFusion Singh et al. (2023) leverage diffusion-based techniques for code generation. These pre-trained models exhibit great potential in code generation tasks, achieving notable improvements in accuracy through various architectures and training strategies. However, they still face challenges in ensuring the syntactical and functional correctness of the generated code.

### 2.2 REINFORCEMENT LEARNING ON CODE

Reinforcement learning (RL) is a method that learns optimal strategies through reward signals from interacting with the environment Fujimoto et al. (2019). It excels in sequence generation tasks, such as enhancing performance in translation and summarization models by improving BLEU and ROUGE scores Ahn et al. (2019). Unlike traditional natural language processing tasks, code generation is more complex. Beyond syntactical correctness, the generated code must be functionally accurate, meaning it should compile and perform as expected in various scenarios. Passing unit tests is important for verifying correctness, safety, and precision, but not sufficient unless the tests are comprehensive and well-designed. Unit tests must cover diverse use cases and edge cases to ensure the code adheres to standards and meets requirements. For instance, the RL-based code generation fine-tuning framework, CodeRL Le et al. (2022), guides fine-tuning by integrating unit test signals with reinforcement learning. PPOCoder Shojaee et al. (2023) improves on CodeRL by employing Proximal Policy Optimization to refine the approach, while RLTF Liu et al. (2023a) incorporates specific error information from the code and multi-granularity feedback alongside an online framework for model training. StepCoder Dou et al. (2024) enhances the code generation process through compiler feedback and segmental optimization. However, current RL-based methods still face some limitations. Primarily, these methods lack detailed feedback on specific types and distributions of programming errors, which is crucial for identifying patterns, understanding root causes, and creating targeted interventions. This makes addressing recurring issues difficult. Additionally, the corrective mechanisms are not robust enough, often failing to provide timely, specific guidance, hindering effective learning. Additionally, the diversity of input-output examples in existing benchmarks is limited, restricting the model's ability to adapt and generalize to different or unseen problems Liu et al. (2023a).

## 3 PROBLEM SETTING

We aim to enhance the automated code generation capabilities of Large Language Models (LLMs) by addressing key challenges in accuracy, diversity, error correction, and code quality. Formally, given a high-level specification $D$, the task is to generate a sequence of code tokens $W = \{w_1, w_2, \ldots, w_T\}$

that maximizes the conditional probability $P(W|D, \theta)$, where $\theta$ represents the model parameters. The optimization objective is defined as $\theta^* = \arg\max_\theta \mathbb{E}_{D \sim \mathcal{D}} [\log P(W|D, \theta)]$.

**Challenges.** Effective code generation is impeded by several factors. **Accuracy** requires minimizing syntactical or logical errors in $W$ to ensure correct functionality. **Diversity** of input-output examples $\mathcal{X}$ is often limited, restricting the model's ability to generalize across varied programming tasks. Efficient **error correction** mechanisms are necessary to identify and rectify errors in $W$, ensuring robust performance. Maintaining high **code quality**, which encompasses adherence to coding standards, style guidelines, and managing code complexity $C(W)$, remains a persistent challenge. Additionally, **adaptability** refers to the model's capacity to adapt to new tasks and incorporate feedback for continuous improvement, which is constrained without robust memory mechanisms.

**FALCON Framework.** To address these challenges, we propose the FALCON framework. The framework leverages both a long-term memory buffer $\mathcal{M}_{\text{long}} = \{(D_i, W_i, T_i, F_i)\}_{i=1}^N$ and a short-term memory buffer $\mathcal{M}_{\text{short}} = \{(D_j, W_j, T_j, F_j)\}_{j=1}^M$ to utilize diverse feedback, enabling fine-tuning of $\theta$. Formally, FALCON seeks to optimize $\theta$ by maximizing a composite reward function $R(W, F) = \alpha T(W) + \beta S(W) + \gamma C(W) + \delta E(W)$, where $T(W)$, $S(W)$, $C(W)$, and $E(W)$ represent unit test, code style, complexity, and error feedback, respectively. The optimization objective is defined as $\theta^* = \arg\max_\theta \mathbb{E}_{D \sim \mathcal{D}} [\mathbb{E}_{W \sim P(W|D, \theta)} [R(W, F)]]$.

**Assumptions.** The effectiveness of FALCON is based on the following assumptions: 1. **Exchangeability** of the dataset $\mathcal{D} = \{D_i\}_{i=1}^N$, implying that the order of tasks does not influence model performance; 2. **Independence** of feedback signals $F$ given the generated code $W$; 3. Sufficient **memory capacity** in both $\mathcal{M}_{\text{long}}$ and $\mathcal{M}_{\text{short}}$ to store relevant interactions without significant data loss; and 4. **Feedback efficacy**, ensuring that integrated feedback mechanisms provide meaningful and actionable information to guide the optimization of $\theta$.

**Optimization Objective.** Given a set of tasks $\mathcal{T} = \{D_i\}_{i=1}^N$, the goal is to learn optimal parameters $\theta^*$ that maximize the expected composite reward across all tasks, formulated as $\theta^* = \arg\max_\theta \frac{1}{N} \sum_{i=1}^N R(W_i, F_i)$, where $W_i \sim P(W|D_i, \theta)$ is the generated code for task $D_i$, and $F_i$ represents the aggregated feedback. The formulation ensures continuous improvement by leveraging both historical and recent feedback to enhance code generation quality.

# 4 METHODOLOGY

In this section, we explore the FALCON framework, which integrates comprehensive unit testing with reinforcement learning, supported by both long-term and short-term memory buffers. During the code generation process, the system stores task descriptions, generated code, and various feedback (e.g., compilation results, code style, and complexity) in the long-term memory buffer. By retrieving this information, the model references high-quality code, avoids past mistakes, and ensures adherence to required standards. After generating the code, a judge model evaluates it and calculates rewards based on the feedback, which are then used to update the model's parameters through reinforcement learning. All generated code and feedback are stored for future reference and optimization. The combination of long-term and short-term memory feedback in the FALCON framework allows the model to not only learn from a wide range of historical data but also adapt quickly to new tasks based on recent performance. The overall framework is illustrated in Figure 1.

## 4.1 TASK DEFINITION: CODE GENERATION BY LLMs

The code generation task involves the automated creation of computer code $W$ from a high-level specification of the desired behavior $D$. The goal of this process is to enhance the likelihood of generating code that effectively solves a given problem, which is represented as a conditional probability $P(W|D)$. This task can be modeled as a sequence-to-sequence problem, aiming to maximize the conditional probability of generating the correct output given the input and the parameters of the LLM model, mathematically represented as:

$$\max_\theta P(W|D, \theta) = \max_\theta \prod_{t=1}^T p(w_t|D, \theta, w_{1:t-1}) \tag{1}$$

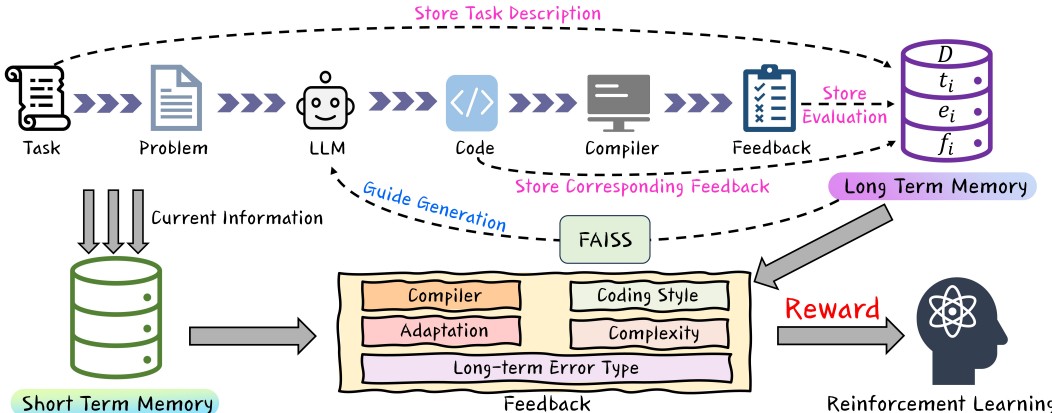

Figure 1: The overview of the FALCON framework. Detailed feedback process (see Appendix Figures 5 and 6 for feedback cases). Reinforcement learning updates using feedback to compute RL loss and optimize the model (see Figure 2 for the detailed meta-reinforcement learning framework).

where $\theta$ denotes the parameters of the LLM model. The LLM model is employed to learn the conditional distribution and is trained on a set of input-output pairs. During the training phase, the parameters $\theta$ are optimized to increase the likelihood of generating accurate outputs for a given input.

### 4.2 MRLF: META-REINFORCEMENT LEARNING WITH DUAL MEMORY BUFFERS

We propose the MRLF algorithm 1, which involves random task sampling from the task distribution during training. Previous works have indicated that different data sampling strategies have varying impacts on model information extraction Zhang et al. (2024). Consequently, we implement both long and short memory sequences. The long memory strategy stores the solutions generated for each problem and the compiler feedback results, whereas the short memory sequence selects the latest samples and unit test feedback from each current iteration. To address repetitive runtime and compilation errors, the long memory strategy categorizes and stores various errors by their types and records the corresponding error lines, enabling fine-grained reward allocation.

---

**Algorithm 1** MRLF Algorithm

---

**Require:** Task distribution $\mathcal{T}$
**Ensure:** Updated model parameters $\theta$
 1: Initialize $LMB, SMB, \theta$
 2: Populate $LMB, SMB$ via few-shot demonstrations
 3: **repeat**
 4:    Sample batch $\{T_i\}$ from $\mathcal{T}$
 5:    **for** each task $T_i$ **do**
 6:       **Code Generation:** Generate code $\hat{W}$ and test results; record in $SMB$
 7:       **Adaptation:** Set $\theta_i = \theta$; sample mini-batch from $LMB, SMB$
 8:       Compute inner loss: $L_{\text{inner}} = \sum L_j$ ($j \in \{\text{sl, coarse, error, complexity, style, negative}\}$)
 9:       Update $\theta_i$: $\theta_i \leftarrow \theta_i - \alpha \nabla_{\theta_i} L_{\text{inner}}(\theta_i)$
10:       **Evaluation:** Assess $\theta_i$ on $T_i$; document outcomes
11:    **end for**
12:    **Meta-Update:** $L_{\text{meta}} = \sum_{i=1}^{N} L_{\text{inner}}(\theta_i)$; refine $\theta$: $\theta \leftarrow \theta - \beta \nabla_\theta L_{\text{meta}}$
13: **until** Convergence
14: **return** $\theta$

---

We initialize two memory buffers (LMB, SMB) and model parameters $\theta$, then randomly sample tasks from the task distribution, executing steps for each one. First, the corresponding code is generated and tested, with results stored in SMB. During adaptation, the algorithm extracts experimental data from

LMB and SMB to aid rapid adaptation to the current task. The inner loss function $L_{\text{inner}}$, incorporating factors like accuracy, complexity, style, and negative examples, is optimized using policy gradient. In evaluation, the optimized parameters are tested, and results are stored for future use. Finally, the losses from all tasks are aggregated to compute the meta-loss $L_{\text{meta}}$, used to update global parameters $\theta$. Algorithm 1 summarizes the framework. For inner loop optimization, our method explores the target space by combining unit test feedback, code complexity, and style norms, using the generated code $\hat{w}$ to construct the reinforcement learning loss function as shown below:

$$L_{r1} = -\sum_{t=S_{\text{fine}}}^{E_{\text{fine}}} R_{\text{fine}}(\hat{w}_t) \log p(\hat{w}_t | D, \theta, \hat{w}_{1:t-1}) \tag{2}$$

where $R_{\text{fine}}(*)$ represents the reward coefficient, and $S_{\text{fine}}$ and $E_{\text{fine}}$ denote the start and end positions of the code snippet, respectively. These values are determined based on different types of feedback. To stabilize the training process, we adopt the supervised learning loss $L_{sl}$ by minimizing the cross-entropy loss as shown below:

$$L_{sl} = -\log P(w \mid D, \theta) = -\sum_{t=1}^{T} \log p(w_t \mid D, \theta, w_{1:t-1}) \tag{3}$$

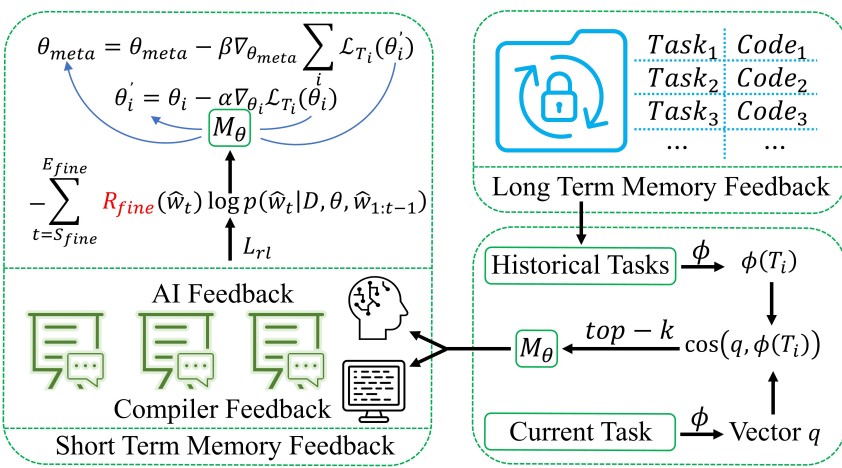

Figure 2: The framework integrates meta-reinforcement learning with both long-term and short-term memory feedback. Long-term memory retrieves historical tasks, while short-term memory provides AI and compiler feedback to refine code generation.

As depicted in Figure 2, we employ a meta-reinforcement learning framework to optimize code generation by integrating both long- and short-term memories for enhanced adaptability. From a global perspective, Long-Term Memory $D_{\text{long}}$ stores historical tasks, generated codes, and feedback to provide valuable context. From a local perspective, Short-Term Memory $D_{\text{short}}$ focuses on recent feedback to enable real-time adjustments. This approach leverages the MAML framework (Finn et al., 2017) for efficient task adaptation with minimal updates.

**Short-Term Memory Adaptation.** Short-Term Memory is utilized to adapt the model locally by adjusting its parameters based on recent feedback. For each task $\mathcal{T}_i$, the inner loop optimization updates the parameters:

$$\theta'_i = \theta_i - \alpha \nabla_{\theta_i} \mathcal{L}_{\mathcal{T}_i}(\theta_i) \tag{4}$$

where $\alpha$ is the learning rate and $\mathcal{L}_{\mathcal{T}_i}$ is the task-specific loss function.

**Global Optimization.** The outer loop performs global optimization of the meta-learning parameters $\theta_{\text{meta}}$ by aggregating feedback across multiple tasks:

$$\theta_{\text{meta}} = \theta_{\text{meta}} - \beta \nabla_{\theta_{\text{meta}}} \sum_i \mathcal{L}_{\mathcal{T}_i}(\theta'_i) \tag{5}$$

where $\beta$ is the meta-learning rate. This ensures better generalization across tasks.

**Final Optimization.** The overall framework combines short- and long-term memory feedbacks with meta-reinforcement learning to achieve coordinated optimization for both global generalization and local task adaptation:

$$\theta_{\text{final}} = \text{Optimize}\left(\theta_{\text{meta}}, \theta, \{\theta_i'\}\right) \tag{6}$$

### 4.3 Long-Term Memory Feedback

Retrieving information from long-term memory significantly improves code quality. We use the FAISS framework (Douze et al., 2024) to retrieve relevant historical code, feedback, and evaluation scores. Task descriptions and feedback are transformed into embedding vectors and then indexed. During code generation, a query vector from the current task retrieves the top-k most similar historical data to guide the process and avoid past errors. The prompt template is provided in the appendix. Consider a set of historical data $\mathcal{D} = (t_i, f_i, e_i)_{i=1}^n$, where $t_i$ represents the task description, $f_i$ is the corresponding feedback, and $e_i$ is the evaluation score. We use an embedding function $\phi(\cdot)$ to transform these tasks and feedback into embedding vectors $\boldsymbol{v}_i = \phi(t_i, f_i)$ and index them with FAISS. During the code generation phase, the current task description $\boldsymbol{t}_{\text{current}}$ and feedback $\boldsymbol{f}$ are transformed into a query vector $\boldsymbol{q} = \phi(\boldsymbol{t}_{\text{current}})$. We compute the similarity between the query vector $\boldsymbol{q}$ and the historical vectors $\boldsymbol{v}_i$ using cosine similarity $\cos(\boldsymbol{q}, \boldsymbol{v}_i)$, and retrieve the top-$k$ most similar historical tasks. The retrieval process can be represented as:

$$\{(t_{i_1}, f_{i_1}, e_{i_1}), \ldots, (t_{i_k}, f_{i_k}, e_{i_k})\} = \text{Top-}k\left(\boldsymbol{v}_i \mid i = 1, 2, \ldots, n\right) \tag{7}$$

By referencing these most relevant historical tasks and feedbacks, the system can guide the current code generation process with past mistakes avoided and ultimate code quality improved.

### 4.4 Short-Term Memory Feedback

During the reinforcement learning phase, we utilize the generated code $\hat{w}$ to construct the reinforcement learning loss function as follows:

$$L_{r1} = -\sum_{t=S_{\text{fine}}}^{E_{\text{fine}}} R_{\text{fine}}(\hat{w}_t) \log p(\hat{w}_t | D, \theta, \hat{w}_{1:t-1}) \tag{8}$$

where $R_{\text{fine}}(*)$ represents the reward coefficient, and $S_{\text{fine}}$ and $E_{\text{fine}}$ denote the start and end positions of the code snippet, respectively. These values are determined based on different types of feedback.
**Compiler Feedback.** For compiler feedback, we adopt the same settings as CodeRL:

$$R_{\text{coarse}}(\hat{W}) = \begin{cases} 1.0, & \text{if } FB(\hat{W}) \text{ is pass} \\ -0.3, & \text{if } FB(\hat{W}) \text{ is failure} \\ -0.6, & \text{if } FB(\hat{W}) \text{ is runtime error} \\ -1.0, & \text{if } FB(\hat{W}) \text{ is syntax error} \end{cases} \tag{9}$$

$$S_{\text{coarse}} = 0, \quad E_{\text{coarse}} = T$$

where $R_{\text{coarse}}$ is based on compiler feedback with the start and end positions set to 0 and $T$.
**Adaptive Feedback.** To enhance the model's efficiency in handling various programming tasks, we devise a mechanism that dynamically adjusts rewards based on the proportion of passes to failures in unit tests. This strategy encourages the model not only to pass unit tests but also to learn from failures, thereby improving its problem-solving capabilities. The reward is calculated as:

$$R_{\text{error}}(\hat{W}) = -0.3 + 1.3 \times \frac{N_{\text{pass}}}{N_{\text{pass}} + N_{\text{fail}}} \tag{10}$$

**Coding Style Feedback.** To further enhance the quality of the generated code, we employ AI Feedback to optimize coding style. An evaluation model scores the generated code based on adherence to the expected coding style standards. The scoring system ranges from -1 to 2, and these evaluation scores are directly used as reward signals in the reinforcement learning process to guide the model toward producing higher-quality code. The coding style assessment template is provided in Table 8.
**Complexity Feedback.** Just like with coding style, we use AI Feedback to evaluate complexity and calculate rewards based on the scores. The complexity assessment template is provided in Table 9.

**Long-term Error Type Feedback.** Introducing a reward mechanism that combines short-term memory recall error rate, current test error, and long-term memory recall of past task performance enables the model to dynamically adjust its rewards based on past error patterns, adapt to various error types and feedback, and ultimately enhance its generalization ability:

$$R_{\text{negative}} = -\sum_{\text{error}} N_{\text{error}} \times P_{\text{error}} \qquad (11)$$

where $N_{\text{error}}$ represents the short-term memory recall, the frequency of each error type in the generated code by the model. This is the immediate feedback for the current task. $P_{\text{error}}$ represents the long-term memory recall, the proportion of each error type. This is the model's performance statistics on long-term tasks, providing a cumulative history of various error types. By correct rewarding, the model can reduce the occurrence of these errors and thereby enhance the accuracy and quality of the generated code.

## 5 EXPERIMENT

### 5.1 QUANTITATIVE EVALUATION ON APPS

To ensure a fair comparison, we use the CodeT5 770M model as our baseline. Our benchmarks include the latest advancements that integrate reinforcement learning (RL) with large language models (LLMs), particularly CodeRL, PPOCoder, and RLTF. For evaluation, we apply the same benchmarks and settings used in these previous works. As shown in Table 1 for the experimental results, our FALCON approach delivers additional performance improvements and surpasses other RL-based methods, indicating that RL with appropriate feedback can effectively improve the model output space and thereby enhance the quality of code generation. In particular, our method achieves the highest pass@1 rates of 8.60%, 2.56%, and 1.25% in the Introductory, Interview, and Competition categories, respectively.

Table 1: Quantitative evaluation on the APPS benchmark. "Intro": introductory, "Inter": interview, "Comp": competition-level tasks. To ensure a fair comparison, we apply these RL-based methods, including PPOCoder, CodeRL, and RLTF, using the same base model, CodeT5, as a backbone. We also compare with models that have a larger number of parameters.

| Method | Size | pass@1 | | | | pass@5 | | | |
|---|---|---|---|---|---|---|---|---|---|
| | | Intro | Inter | Comp | All | Intro | Inter | Comp | All |
| Codex | 12B | 4.14 | 0.14 | 0.02 | 0.92 | 9.65 | 0.51 | 0.09 | 2.25 |
| GPT-Neo | 2.7B | 3.90 | 0.57 | 0 | 1.12 | 5.50 | 0.80 | 0 | 1.58 |
| CodeT5 base | 770M | 3.85 | 0.58 | 0.02 | 1.05 | 8.52 | 1.53 | 0.25 | 2.82 |
| PPOCoder | 770M | 4.06 | 0.79 | 0.15 | 1.32 | 9.97 | 2.06 | 0.70 | 3.37 |
| CodeRL | 770M | 7.08 | 1.86 | 0.75 | 2.69 | 16.37 | 4.95 | 2.84 | 6.81 |
| RLTF | 770M | 8.40 | 2.28 | 1.10 | 3.27 | 18.60 | 5.57 | **3.70** | 7.87 |
| **Ours** | 770M | **8.60** | **2.56** | **1.24** | **3.50** | **19.75** | **5.85** | 3.57 | **8.17** |

### 5.2 QUANTITATIVE EVALUATION ON HUMANEVAL AND MBPP

To further validate the effectiveness of our method, we evaluate the zero-shot performance of the DeepSeek-Coder-Instruct model, trained with our method on our custom dataset, using the well-established MBPP and HumanEval benchmarks. We also compare these results against other reinforcement learning methods, such as PPOCoder and RLTF. The experimental results are illustrated in Table 2.

Table 2: The results of pass@1 on the MBPP and HumanEval benchmarks.

| Model | Humaneval | MBPP |
|---|---|---|
| DeepSeek-Coder-Instruct | 73.8 | 74.9 |
| PPOCoder | 76.8 | 76.2 |
| RLTF | 76.8 | 75.9 |
| **Ours** | **82.9** | **80.7** |

Compared to other reinforcement learning methods, our method consistently achieves the best performance on both the HumanEval and MBPP benchmarks. The significant advantage of our method can be attributed to its diversified feedback mechanism. Unlike other methods that may focus on a single metric, our method continuously optimizes the model's generation capability through multi-dimensional feedback. This approach demonstrates a strong ability to enhance the generation of correct code and proves particularly effective in complex tasks.

### 5.3 QUANTITATIVE EVALUATION ON CODAL-BENCH

In addition to evaluating the functional correctness of the code, we adopt CODAL-Bench, a rigorous and comprehensive benchmark for LLM consistency in coding preferences to validate the effectiveness of short-term memory feedback. DeepSeek-Coder-Instruct-6.7B model is used and the results are illustrated in Figure 3. It is found that there is a noticeable improvement in various coding preferences, particularly in Code Complexity and Coding Style after implementing the FALCON framework. This observation is attributed to the inclusion of feedback on these aspects in the short-term memory. However, the improvement in Instruction Following is not as significant.

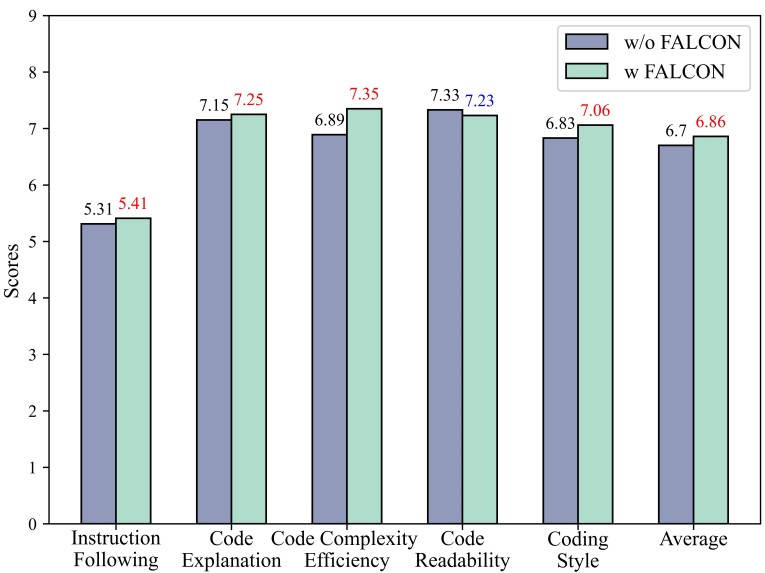

Figure 3: Quantitative evaluation on CODAL-Bench

### 5.4 QUANTITATIVE EVALUATION ON SCICODE

To validate the general-purpose task capabilities of our framework, we also select the SciCode benchmark, which covers challenging research-level coding problems across natural sciences, including mathematics, physics, chemistry, and biology. SciCode decomposes the main problems into several subproblems, making it a particularly rigorous benchmark of a model's coding capabilities. Even the most advanced models nowadays, such as Claude 3.5-Sonnet and ChatGPT-4.0, can only solve 1.5% and 4.6% of the main problems, respectively. Although Deepseek-Coder-6.7B-instruct initially demonstrates a low task pass rate on this benchmark, we observe significant performance improvements on the subproblems after applying our framework due to the utilization of long-term memory mechanisms.

### 5.5 QUANTITATIVE EVALUATION ON AGENTBENCH

To further evaluate the performance of our framework, we conduct a systematic assessment on AgentBench Liu et al. (2023b), focusing specifically on long-term memory capabilities. Since our primary focus is on code generation tasks, we select three environments within AgentBench: Operating System (OS), Database (DB), and Knowledge Graph (KG). In this evaluation, we compare

Table 3: The results of pass@1 on SciCode benchmarks with and without FALCON.

| Model | Size | Method | Subproblem | Main Problem |
|-------|------|--------|------------|--------------|
| CodeLlama | 70B | - | 10.4 | 0 |
| CodeLlama | 7B | w/o | 0.4 | 0 |
| CodeLlama | 7B | w | 3.5 | 0 |
| DeepSeek-Coder | 6.7B | w/o | 5.2 | 0 |
| DeepSeek-Coder | 6.7B | w | 8.3 | 0 |

Table 4: Test set results of AGENTBENCH.

| Model | Size | VER | OS | DB | KG |
|-------|------|-----|-----|-----|-----|
| GPT-3.5-turbo | - | 0613 | 31.6 | 15.7 | 25.9 |
| GPT-4 | - | 0613 | **42.4** | **32.0** | **58.8** |
| Tinyllama | 1.1B | - | 2.8 | 0 | 0 |
| Codellama | 7B | instruct | 9.7 | 2.7 | 0 |
| Qwen | 7B | chat | 12.5 | 13.0 | 7.0 |
| Agentlm | 7B | chat | 14.6 | **33** | 9.0 |
| Deepseek-Coder | 6.7B | instruct | 17.4 | 23.3 | 6.8 |
| DeepSeek-Coder (FALCON) | 6.7B | instruct | **22.2** | 26.7 | **9.0** |

proprietary models (such as GPT-4 and GPT-3.5) with open-source models (such as Codellama and Qwen 2.5) Liang et al. (2024). The results reveal that the models optimized through our framework exhibit significant improvements, particularly in the OS environment with an increase of 4.8 percentage points. The experimental results are illustrated in Table 4.

## 5.6 ABLATION STUDIES

**The Influence of Models.** To validate the scalability and robustness of our framework, we conduct experiments with the larger model, DeepSeek-Coder-Instruct-6.7B, to further evaluate its performance. Notably, the improvements in introductory-level tasks are significant, which can be attributed to the use of long-term memory that enhances the quality of generated data and further unlocks the model's potential. The results are illustrated in Table 5.

**The Influence of Different Feedbacks on Coding Preferences.** As shown in Figure 4, ablation experiments are also conducted to validate the effectiveness of the feedback that we introduce for coding preferences. It is found that incorporating targeted feedback enhances the model's performance concerning its respective coding preferences. Notably, the optimization aiming at increasing code complexity achieves the best results. Although there are some improvements in coding style and instruction following, it is worth noting that the enhancement in code instruction following is not particularly significant, suggesting it as a topic for future investigation.

Table 5: Different large language models as the backbone

| Model | Size | Method | Intro | Inter | Comp | All |
|-------|------|--------|-------|-------|------|-----|
| CodeT5 | 770M | w/o | 3.85 | 0.58 | 0.02 | 1.12 |
| CodeT5 | 770M | w | 8.60 | 2.56 | 1.25 | 3.50 |
| DeepSeek-Coder | 6.7B | w/o | 16.70 | 7.20 | 2.30 | 8.12 |
| DeepSeek-Coder | 6.7B | w | 22.40 | 8.52 | 3.70 | 10.33 |

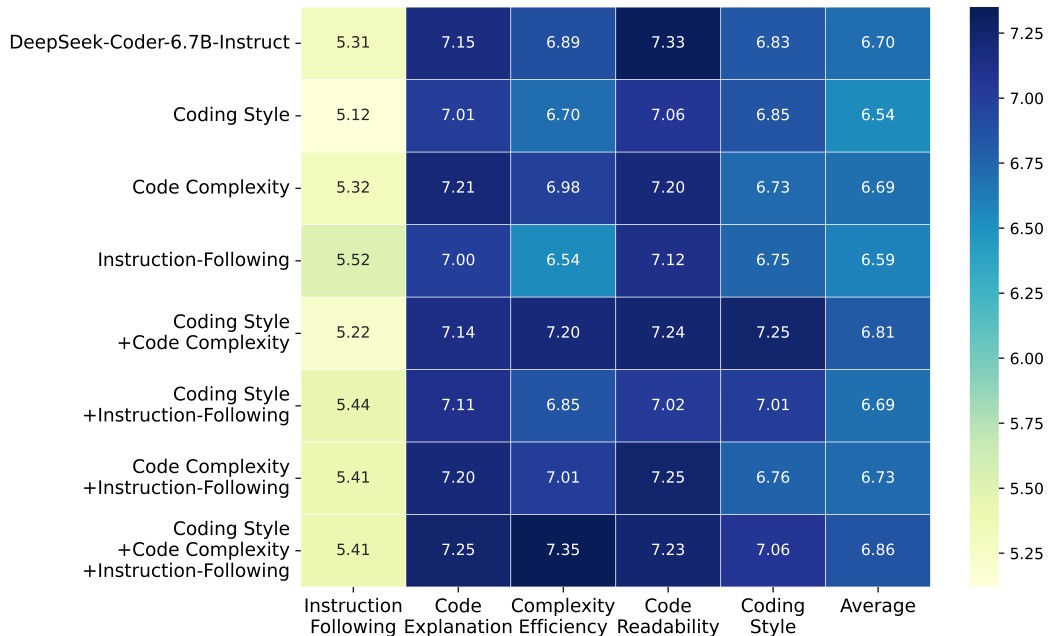

Figure 4: Effect of different feedbacks on coding preferences

**The Influence of Memory.** To validate the impact of long-term and short-term memories on code generation capabilities, we conduct ablation experiments using CodeT5 as the base model and test it on the APPS dataset. As shown in Table 6, the experimental results indicate that both long- and short-term memory feedbacks enhance the model's code generation performance effectively, while the short-term memory feedback demonstrates a more significant improvement. This improvement can be attributed to the effective reward design which plays a positive role in fine-tuning the model.

Table 6: Effect of long and short memories on different performance metrics

| Long Memory | Short Memory | Intro | Inter | Comp | All |
|:---:|:---:|:---:|:---:|:---:|:---:|
| - | - | 3.85 | 0.58 | 0.02 | 1.12 |
| ✓ | - | 4.14 | 0.74 | 0.02 | 1.28 |
| - | ✓ | 7.20 | 1.86 | 0.70 | 2.70 |
| ✓ | ✓ | 8.60 | 2.56 | 1.25 | 3.50 |

## 6 CONCLUSIONS AND FUTURE WORK

In this work, we propose **FALCON**, a novel framework that enhances automated code generation by integrating long-term and short-term memory feedbacks within a meta-reinforcement learning strategy. Long-term memory retains past interactions to improve code quality and reduce repetitive mistakes, while short-term memory enables immediate adjustments based on recent feedback from compilers and AI systems. This dual-memory approach addresses limitations in existing models that struggle to align code generation with user intent, especially in specialized tasks or edge cases. By incorporating non-differentiable code features like style and complexity into the feedback loop, FALCON ensures that the generated code is not only functionally correct but also adheres to real-world programming standards. Extensive evaluations on benchmarks including APPS, HumanEval, and CODAL-Bench demonstrate that FALCON outperforms existing RL-based methods, achieving higher functional correctness and better coding style adherence. In future work, we aim to expand FALCON's capabilities by incorporating a broader diversity of programming languages and tackling more complex code generation challenges.

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

## A EXAMPLE

We provided an example of code generation with long-term memory. When generating code without long-term memory, it often results in repetitive ValueError issues. By incorporating long-term memory to retrieve the most relevant code blocks and embedding them as context during generation, the quality of the generated code can be significantly improved. We also provided an instance of using long-term memory in SCICode. In direct generation, there were certain logical issues within the code, and the coding style lumped all formulas together. However, after employing long-term memory retrieval for assistance, the code was segmented appropriately and the logic was correctly implemented.

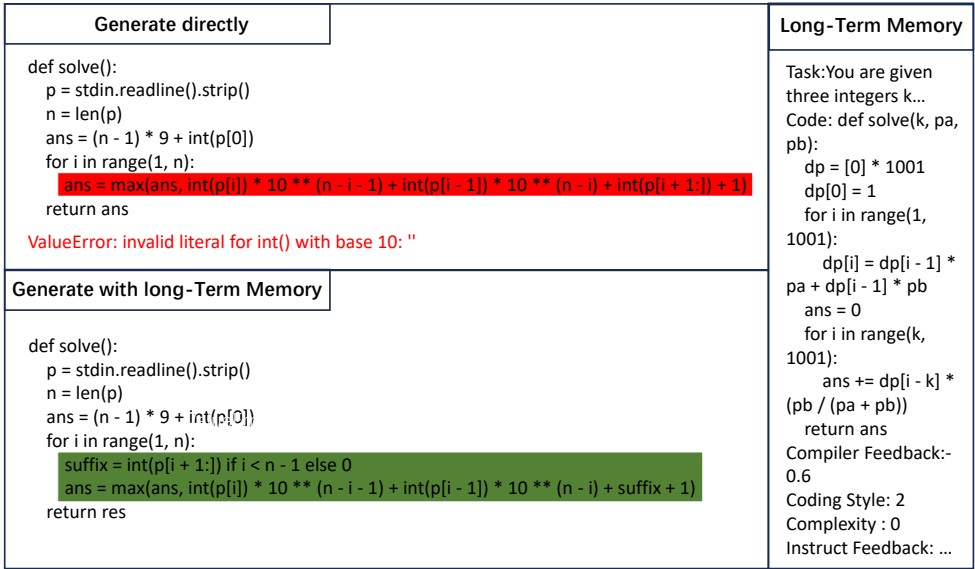

Figure 5: An example of code generation with long-term memory incorporated.

## B FEEDBACK ERROR CORRELATION ANALYSIS

We performed an analysis of different feedback types and their associated error categories. We conducted experiments using CodeT5 as the base model on APPS with individual feedback and collected the occurrence frequency of various sub-errors. The results are shown in the Figure 7. The experimental results indicate that compiler feedback significantly reduces Syntax Errors and Index Errors. However, it also slightly increases the occurrence of Value Errors. This can be attributed to the corrective nature of compiler feedback on errors. Other feedback types, such as Coding Style, Instruction Following, and Code Complexity Feedback, can reduce Syntax Errors compared to having no feedback. However, their reduction is not as significant as that achieved by compiler feedback. Instruction Following Feedback specifically shows some improvement in reducing Value Errors, indicating an enhancement in instruction adherence.

## C PROMPTS

We have compiled relevant prompt templates for code generation based on long-term memory retrieval and AI feedback.

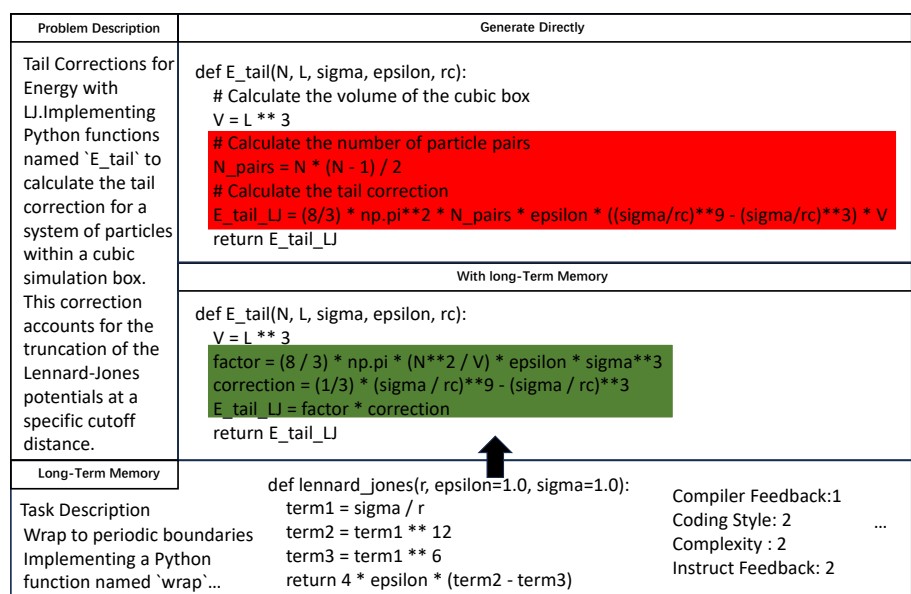

Figure 6: An example of code generation with long-term memory incorporated.

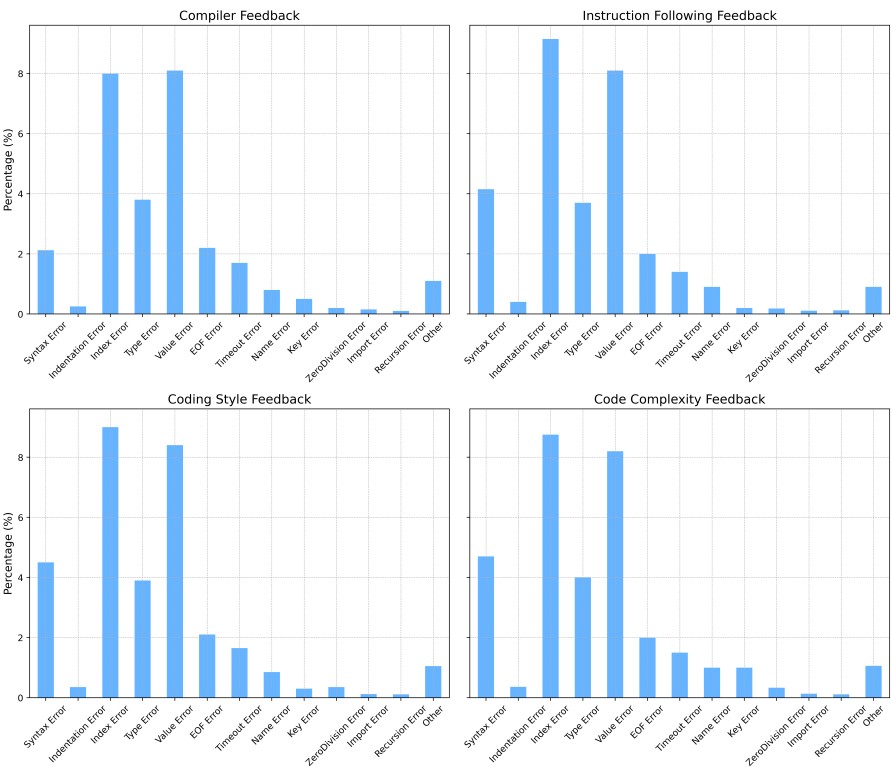

Figure 7: Frequency of sub-errors generated by different feedback types.

Table 7: Code generation template

| **Instruction:** |
| Please write a Python function based on the task description, referencing the historical context for inspiration. Ensure that the generated code follows the provided requirements and avoids the listed errors. |
| **Task:** [instruction] |
| **Context of relevant code:** |
| - **Historical Task:** [Brief description of a similar task] |
| - **Code:** [Code snippet] |
| - **Style Score:** [Style score] |
| - **Efficiency Score:** [Efficiency score] |
| - **Additional Feedback:** [Additional comments or issues] |
| **Requirements:** |
| 1. Ensure the generated code adheres to best practices for Python, including proper naming conventions, consistent formatting, and coding standards. |
| 2. Optimize the code for performance, avoiding unnecessary recursion or nested loops. |
| 3. Use built-in or efficient library functions whenever applicable to improve both readability and performance. |
| **Avoid the following errors:** |
| - [Historical error] – Avoid structural or logical issues found in previous code snippets. |
| **Output:** Ensure your response is in the format of '''python'''. |

Table 8: Coding style assessment template.

| **Coding Style Assessment** |
| Evaluate the coding style of provided code segments. Assess how well the code adheres to the best practices of the language, focusing on readability, maintainability, and efficiency in line with the language's idiomatic style. |
| **Reward Scale: Rate outputs on a scale of -1 to 2** |
| **-1. Poor Adherence:** The code significantly deviates from standard practices, showing poor readability, maintainability, and efficiency. |
| **0. Basic Adherence:** The code makes some effort to follow language conventions but lacks consistency in readability, maintainability, or efficiency. |
| **1. Good Adherence:** The code generally follows standards, demonstrating adequate readability, maintainability, and efficiency, though with room for improvement. |
| **2. Excellent Adherence:** The code exemplifies best practices, with high readability, maintainability, and efficiency, fully adhering to idiomatic conventions. |

## D   ERROR CATEGORY

Due to the differences in languages accepted by Compiler Feedback during unit tests for various language tasks, we have standardized the definition of sub-errors in Compiler Feedback. The table 11 12 13 below outlines our specifications for Python, C, and Java.

Table 9: Complexity assessment template.

| Complexity Assessment |
|---|
| Evaluate the solutions and code provided by the assistant based on their complexity. Assess how well the code manages complexity while achieving the desired outcomes. |
| **Reward Scale: Rate outputs on a scale of -1 to 2** |
| **-1. Overly Complex:** The code is unnecessarily complicated, with a high level of complexity that makes it hard to understand or maintain. |
| **0. Acceptable Complexity:** The code has a reasonable level of complexity, but there may be opportunities to simplify further. |
| **1. Moderately Simple:** The code is simple and well-organized, with minimal complexity and clear logic. |
| **2. Optimal Simplicity:** The code exemplifies the best practices in minimizing complexity while ensuring functionality. |

Table 10: Instruction following assessment template.

| Instruction Following Assessment |
|---|
| Evaluate the assistant's fidelity to provided instructions. Assess how accurately the assistant's responses align with user directives, noting any deviations and their justification. |
| **Evaluation Criteria** |
| *Precision in Following Instructions*: Does the assistant adhere to the specifics of the provided instructions? |
| *Justification for Deviations*: If deviations occur, are they justified by critical necessity or explicit user request? |
| *Alignment with User Directives*: How well do the assistant's responses match the user's specified needs and expectations? |
| *Necessity of Deviations*: Are any deviations from instructions made only in situations deemed absolutely necessary or upon direct user request? |
| **Reward Scale: Rate outputs on a scale of -1 to 2** |
| **-1. Non-Compliant:** The assistant frequently deviates from instructions without necessity or user consent. |
| **0. Acceptable:** The assistant shows some adherence to instructions but deviates without strong justification. |
| **1. Compliant with Justified Deviations:** The assistant generally follows instructions, with deviations occurring but justified by necessity or user request. |
| **2. Fully Compliant:** The assistant follows instructions closely, with minimal deviations, all of which are well justified. |

Table 11: Common Python errors with categories.

| Sub-error | Description | Category |
|---|---|---|
| Syntax Error | Code contains syntax errors that cause the compilation to fail. | Syntax Error |
| Indentation Error | Wrong indentation format. | Syntax Error |
| Index Error | Index operation is out of bounds. | Runtime Error |
| Type Error | An operation or function was applied to an object of an inappropriate type. | Runtime Error |
| Value Error | An operation or function received an argument with the correct type but with an inappropriate value. | Runtime Error |
| EOF Error | The input() function encountered an end-of-file condition (EOF) without reading any data. | Runtime Error |
| Timeout Error | Code execution time exceeds time limit. | Runtime Error |
| Name Error | A local or global name is not defined. | Runtime Error |
| Key Error | A mapping (dictionary) key is not found in the set of existing keys. | Runtime Error |
| Import Error | The imported package is not found. | Runtime Error |
| ZeroDivision Error | The second argument of a division or modulo operation is zero. | Runtime Error |
| Recursion Error | Code execution recursive operation exceeds the maximum limit. | Runtime Error |

Table 12: Common C Language errors with categories.

| Sub-error | Description | Category |
|---|---|---|
| Segmentation Fault | Accessing memory that the program doesn't have permission to access. | Runtime Error |
| Null Pointer Dereference | Attempting to dereference a pointer that is NULL. | Runtime Error |
| Buffer Overflow | Writing data outside the allocated buffer memory. | Runtime Error |
| Memory Leak | Dynamically allocated memory not being freed. | Runtime Error |
| Syntax Error | A syntax mistake in the code, such as a missing semicolon. | Syntax Error |
| Type Mismatch | Assigning a value of one type to a variable of another type. | Syntax Error |
| Uninitialized Variable | Using a variable before it has been initialized. | Runtime Error |
| Undefined Behavior | Code that can exhibit unpredictable behavior depending on compiler or runtime environment. | Runtime Error |
| Division by Zero | Attempting to divide a number by zero. | Runtime Error |
| Infinite Loop | A loop that never terminates due to incorrect logic. | Runtime Error |

Table 13: Common Java Language errors with categories.

| Sub-error | Description | Category |
|---|---|---|
| NullPointerException | Attempting to access an object with a null reference. | Runtime Error |
| ArrayIndexOutOfBounds-Exception | Accessing an array index that is out of bounds. | Runtime Error |
| ClassCastException | Casting an object to a subclass it is not an instance of. | Runtime Error |
| NumberFormatException | Attempting to convert a string to a number, but the string doesn't have the appropriate format. | Runtime Error |
| StackOverflowError | Recursive method calls that exceed the stack size. | Runtime Error |
| Syntax Error | Any mistake in the code structure such as missing braces or semicolons. | Syntax Error |
| ClassNotFoundException | The Java class is not found at runtime. | Runtime Error |
| IllegalArgumentException | A method has been passed an illegal or inappropriate argument. | Runtime Error |
| ArithmeticException | Division by zero or other illegal arithmetic operations. | Runtime Error |
| UnsupportedOperation-Exception | When a requested operation is not supported. | Runtime Error |

