# OpenReview forum: "FALCON: A Feedback-Driven Adaptive Long/Short-Term Memory Reinforced Coding Optimization"
_ICLR.cc/2025/Conference — ICLR 2025 Conference Withdrawn Submission_

### Official Review · Reviewer_8qfG · 2024-10-29

**Soundness:** 3
**Presentation:** 2
**Contribution:** 3
**Rating:** 3
**Confidence:** 4

**Summary:**

This paper introduces FALCON, an innovative framework that enhances automated code generation by combining long-term and short-term memory feedback within a meta-reinforcement learning strategy. Long-term memory captures previous interactions to improve code quality and minimize repetitive errors, while short-term memory allows for immediate adjustments based on recent feedback from compilers and AI systems. This dual-memory approach overcomes limitations in existing models that struggle to align with user intent, particularly in specialized tasks or edge cases. By integrating non-differentiable code features such as style and complexity into the feedback loop, FALCON ensures that generated code is not only functionally correct but also meets real-world programming standards. Extensive evaluations on benchmarks like APPS, HumanEval, and CODAL-Bench show that FALCON outperforms current RL-based methods, achieving higher functional accuracy and better adherence to coding style.

**Strengths:**

The paper introduces FALCON, an innovative framework for automated code generation that combines long-term and short-term memory with meta-reinforcement learning. And it incorporates non-differentiable features like coding style and complexity into reinforcement learning loops, ensuring that the generated code is not only functionally correct but also adheres to real-world programming standards. FALCON demonstrates state-of-the-art performance on benchmarks including APPS, HumanEval, and CODAL-Bench, outperforming existing RL-based methods.

**Weaknesses:**

The main issue with this paper is the lack of clarity and solid foundation in the Problem Setting and Methodology sections. For instance, in lines 125-130, the authors present four assumptions for the effectiveness of FALCON without justifying why these assumptions hold. From my perspective, none of these assumptions can be easily satisfied. Specifically, Assumption 1 suggests that task order doesn’t affect model performance, yet models typically are more influenced by recently seen examples. Additionally, feedback signals often overlap, and the proposed feedback rewards seem too coarse-grained to provide meaningful guidance for optimizing model parameters. There are also inconsistencies and undefined symbols (like the T and F in the FALCON Framework subsection) in the formulas, leading to confusion. For example, the optimization objective given in lines 131-135 differs from that given in the FALCON Framework subsection.

**Questions:**

1. In Section 4.3, you mention that you retrieve the top-$k$ most similar historical tasks. But you do not explain how these historical tasks can be used to provide long-term memory feedback in detail. Could you elaborate on this?
2. In the Assumptions subsection, you propose four assumptions for FALCON's effectiveness. How do you ensure these assumptions are satisfied? Could you provide empirical evidence or theoretical justification for each assumption, particularly the task order independence? Also, it would be beneficial if you could elaborate on how to handle potential overlap in feedback signals and how the feedback rewards are designed to be sufficiently fine-grained for effective parameter optimization.
3. Could you provide more details about the reward coefficient in Equation 8? It seems some information is missing.

---

### Official Review · Reviewer_F4PX · 2024-11-01

**Soundness:** 2
**Presentation:** 1
**Contribution:** 1
**Rating:** 1
**Confidence:** 4

**Summary:**

This paper describes a reinforcement learning framework to fine-tune large language models (LLMs) for code generation. In particular, they employ meta reinforcement learning (MAML), which is traditionally used to train models that can be quickly fine-tuned for new tasks, to arrive at a model that is already fine-tuned to solve new tasks.

**Strengths:**

I think the general idea of applying meta-RL to the LLM domain is interesting, and the main subject (code generation with LLMs) is timely and of importance.

**Weaknesses:**

The paper in its current form is very hard to read and understand. I would propose a major rewrite of the paper, focusing on structure/clarity (e.g., after the introduction I still had no idea what the method would be), content (the are many statements that are pure and vague speculation. e.g., L22 or L227 to pick just two examples) as well as grammar/style (e.g., past tense in abstract to the describe the current state of affairs, introducing an abbreviation (CLM, L73) that is never used again).

The motivation of the method is not clear. What I found confusing:
- In section 3 you say that the overall optimization problem for code generation is to find the most likely completion under the model. This is the standard objective for LLM training, but presumably you care about correctness of the code here, no?
- Long- vs short-term memory. There is much touting on the benefits, but what actually happens is revealed pice-meal and annoying to find out (confusions remain; see below)
- It's not clear to me why you think that MAML will help you here. Its motivation is to make adaptation to new tasks easier, but that's not your use-case anyway, right?

It's hard to draw any insights from the experiments. Major points that need to be addressed:
- State clearly what data you train on for each experiment, and what the baselines are
- Start training with an up-to-date model; the CodeRL/PPOCoder etc. results are outdated by now and their numbers are very low compared to current models
- If you bring your own dataset, you need to provide at least a SFT baseline with it.
- A baseline with "standard" RL on your dataset, i.e., without MAML.

Finally, the paper omits any hyper-parameters, e.g., the coefficients for the different reward terms, let alone optimization-related HPs such as batch sizes. Reproducibility is this very low and it's hard to judge the reported results.

**Questions:**

At this point it is still not clear to me what the role of short- and long-term memory buffers are. According to Algorithm 1 it just contains few-shot samples? Is the long-term buffer used during inference? Some experimental results suggest so but it's never mentioned.

---

### Official Review · Reviewer_ABHj · 2024-11-01

**Soundness:** 3
**Presentation:** 2
**Contribution:** 2
**Rating:** 5
**Confidence:** 4

**Summary:**

This paper presents a long/short-term memory framework for code generation. This framework defines four rewards according to correctness, code style, complexity, and error type stats. The authors jointly optimize them using the MAML framework with policy gradient loss and SFT loss. The paper presents experiments on several benchmarks to show the effectiveness of the proposed framework.

**Strengths:**

I see 3 main contributions from the framework presented by the paper:

1. A *combination* of RAG and MAML and application for code generation.
2. The combination of 4 rewards beyond functional correctness to put into the RL pipeline.
3. Empirically, comprehensive and solid experiments on various benchmarks.

In detail, experiments (with CodeT5 770M model and DeepSeek Coder 6.7B Instruct) are conducted on multiple benchmarks (reporting pass@1/pass@5): APPS, HumanEval, MBPP, CODAL-Bench, SciCode, and AgentBench to validate the effectiveness of the proposed framework. I especially appreciate the evaluation of CODAL-Bench, which shows metrics beyond functional correctness and a complete set of ablation of different rewards within this MAML framework to show the individual effect of each task/reward.

**Weaknesses:**

### Method

1. Complementary to what I mention in the Strength part, what I see as novel is the *combination* of RAG and MAML, while each component seems to be directly applied in this work:

  1.1 The long-term memory presented in the paper: finding the Top-K most similar data points and putting them into the prompt context, resembles the retrieval augmented generation (RAG). However, I didn’t find any literature review in this paper that covers this topic, which has also been used in code generation, such as the classic code generation and summarization[1], or the recently popular code agent domains like SWE-Bench[2]. It would be great for the authors to discuss at least this topic and clarify whether there's a difference in the methods presented or it's a direct application to code generation.

  1.2. The proposed MRLF (Sec 4.2, algorithm 1) seems like a direct application of MAML (Finn et al. 2017, algorithm 3 for Reinforcement Learning) to code generation with limited algorithmic difference/novelty on top of MAML.

2. The defined rewards are a weighted sum of 4 individual rewards (L122), which might require a heavy hyperparameter sweep/tuning.

### Readability & Presentation

The manuscript could benefit from more revision in terms of presentation, especially the formula and symbol used. Here are what I spot but not limited to:

1.	There is no pointer to Table 3 in the main text.
2.	Eq(2) and Eq(8) are the same.
3.	L109: $\mathcal{D}$ shows up for the first time but not defined until L127. Please move the def ahead.
4.	L113: $\mathcal{X}$ defined but never used in the paper.
5.	L119: I'm confused by the $F_i$ here. Is it the same as $F$ defined later in L128? If so, please move the def ahead.
6.	L119: I'm confused by the $T_i$ here? It appears in multiple places: L22 for $T(W)$, algorithm 1 line 4 for task $T_i$, and L310 for token length. Could you use different variables for these separate meanings to avoid collision?
7.	L122: The reward definition $R(W, F)$ depends on $F$ but it’s RHS does not seem to depend on $F$?


[1] Retrieval Augmented Code Generation and Summarization https://arxiv.org/abs/2108.11601
[2] SWE-Bench https://arxiv.org/pdf/2310.06770

**Questions:**

1. Could the author clarify the algorithmic difference between the proposed MRLF (Sec 4.2, algorithm 1) and the algorithm in the MAML framework (Finn et al. 2017, Algorithm 1 and Algorithm 3 for Reinforcement Learning)? Although it is referenced in L258 near the end of Sec 4.2, I want to be clear whether it’s a direct application of MAML in the code generation task (if so, I would appreciate it if the author made this clear at the beginning of Sec 4.2) or is there algorithmic novelty on top of MAML?

2. How is the long-term memory collected? Is it collected on the fly and grows this long-term memory bank during the evaluation of pass@1, or is it a separate dry-run pass to collect the feedback and another pass to retrieve the Top-K?

3. I would appreciate the clarification of the technical terms “task” and “dataset”: L126 defines datasets $\mathcal{D} = \{ D_i \}$, while L132 also defines tasks $\mathcal{T} = \{D_i\}$. Are they referring to the same thing? This causes confusion because in algorithms 1 Line 4, it says, "sample task $T_i$ from $\mathcal{T}$. Is $T_i$ here the same as $D_i$? This leads to my question about the understanding of your framework: do you treat “task” as each problem instance, e.g. problem statement in APPS, or do you treat “task” as each benchmark? In exp, do you train on all benchmarks: APPS, HumanEval, MBPP, CODAL-Bench, SciCode, and AgentBench, and the sample task $T_i$ in algorithm 1 Line 4 is sampling, for example, APPS, SciCode} and train on these two benchmarks of all problems in the inner loop or it is sampling just problem instance and the MAML is carried out individually on each benchmark?

4. Eq(2) and Eq(8) seem to suggest that the paper uses fine-grained reward $R_{\text{fine}}$ for the token-level reward, while all the reward terms I’ve seen, Eq(9) (10) (11) are answer-level reward. Could the author clarify that throughout this paper, it defaults to the answer-level reward where the reward is placed at the last token generated, and all the other non-terminal tokens are of reward 0?

5. When you combine the SFT loss and the PPO loss, do you have a weight term to balance the 2 losses?

6. The authors use different notations, $\omega$ for Eq(3) and $\hat{\omega}$ for Eq(2) to denote the code. Does this imply that the codes used for PPO loss is generated on-policy while the ones used in SFT loss are off-policy (e.g. coming from the training set)?

7. What is the embedding function $\phi$ used in L282?

8. Understanding the ablation in Table 6: do the 2 rows without short-term memory report performance without any training, while the last 2 rows are with training?

9. Could the author report the hyperparameters, esp. those weighted term for rewards used in the experiments?

---

### Note · Authors · 2024-11-21

I have read and agree with the venue's withdrawal policy on behalf of myself and my co-authors.